# Genomic Variation-Mediating Fluconazole Resistance in Yeast

**DOI:** 10.3390/biom12060845

**Published:** 2022-06-17

**Authors:** Wen-Yao Wang, Hong-Qing Cai, Si-Yuan Qu, Wei-Hao Lin, Cheng-Cheng Liang, Hao Liu, Ze-Xiong Xie, Ying-Jin Yuan

**Affiliations:** Frontiers Science Center for Synthetic Biology, Key Laboratory of Systems Bioengineering (Ministry of Education), School of Chemical Engineering and Technology, Tianjin University, Tianjin 300072, China; 2021207664@tju.edu.cn (W.-Y.W.); chq325@tju.edu.cn (H.-Q.C.); sh_qsy@tju.edu.cn (S.-Y.Q.); 3019207286@tju.edu.cn (W.-H.L.); lcc1588_@tju.edu.cn (C.-C.L.); leo3019207284@tju.edu.cn (H.L.); yjyuan@tju.edu.cn (Y.-J.Y.)

**Keywords:** antifungal resistance, fluconazole, yeast, genomic variations, aneuploidy, loss of heterozygosity

## Abstract

Fungal infections pose a serious and growing threat to public health. These infections can be treated with antifungal drugs by killing hazardous fungi in the body. However, the resistance can develop over time when fungi are exposed to antifungal drugs by generating genomic variations, including mutation, aneuploidy, and loss of heterozygosity. The variations could reduce the binding affinity of a drug to its target or block the pathway through which drugs exert their activity. Here, we review genomic variation-mediating fluconazole resistance in the yeast *Candida*, with the hope of highlighting the functional consequences of genomic variations for the antifungal resistance.

## 1. Introduction

Nearly 1 billion people suffer from fungal infections, and fungal diseases caused over 1.6 million deaths annually, and more than 10 million people have mucosal candidiasis [1,2]. The yeast *Candida* is one of the most frequent fungal pathogens, including *Candida albicans*, *Candida tropicalis*, *Candida parapsilosis*, *Candida auris,* and *Candida glabrata*. The yeast *Candida* is an opportunistic pathogen present in the normal microbiome, which can overgrow in immune-compromised individuals and cause disseminated infections [1,3]. Available antifungal drugs are limited, and fluconazole is one of the most widely used antifungal agents [4]. Fluconazole is an efficient inhibitor of lanosterol 14-α-demethylase, which functions to convert lanosterol to ergosterol. Subsequently, the synthesis of normal sterols is blocked, and toxic 14-α-methyl sterols are accumulated in fungi, which is responsible for the fungistatic activity [3,5].

However, with the long-term use of low-concentration antifungal drugs, some types of fungi have become resistant to undergoing abundant genomic variations, including point mutations, aneuploidy, and loss of heterozygosity (LOH). Approximately 0.5–2% of *C. albicans* isolates are resistant to fluconazole, while the proportion of resistant *C. tropicalis*, *C. parapsilosis,* and *C. glabrata* amounts to 4–9%, 2–6%, and 11–13%, respectively [6]. Genomic variations usually lead to increased drug efflux and decreased affinity between the drug and the target. A better understanding of the antifungal resistance mechanism is critical to developing better therapeutics and to improving strategies that may overcome resistance [7]. Here, we focus on fluconazole resistance in yeast pathogens, highlighting the resistance mediated by genomic variations.

## 2. Antifungal Resistances Mediated by Genetic Mutations

Slight alterations in DNA nucleotides in necessary genes can enable notable antifungal resistance. The ergosterol biosynthesis pathway is a complex route involving about 20 enzymes, and most of these enzymes are encoded by a series of *ERG* genes. Mutations in *ERG* genes could affect ergosterol synthesis and lead to fluconazole resistance. Interestingly, functional mutations are mainly represented in *ERG3* and *ERG11*, encoding C-5 sterol desaturase and sterol 14-demethylase, respectively. Additionally, the facilitator superfamily transporter (MFS-T) and the ATP-binding cassette transporter (ABC-T) are also responsible for the fungistatic activity of fluconazole. The MFS-T is encoded by *MDR1,* and the ABC-T is encoded by *CDR1*/*CDR2*. While functional mutations are represented in *MDR1* or *CDR1*/*CDR2*, the overexpression of efflux pumps can also lead to fluconazole resistance (Figure 1A,B) [3].

### 2.1. Genetic Mutations Lead to Abnormal Synthesis of Sterols

The synthesis of ergosterol can be affected by *ERG* mutations. Two mutations, Y132F and K143R, in *ERG11* affect the catalytic efficiency of Erg11. Therefore, the binding affinity of fluconazole to its target is reduced. The isolate with these mutations exhibited a 16-fold increase in fluconazole minimum inhibitory concentration (MIC) [8]. Missense mutations and silent point mutations in *ERG3* are thought to be responsible for the mis-synthesized sterols and high levels of drug resistance [9]. A *C. albicans* isolate containing T51C, T434C, and C1052T substitutions in *ERG3* exhibits an increased fluconazole resistance, with ergosterol substituted by ergosta-7,22-dienol (Figure 1A) [9]. In addition, clinical *Candida* isolates with *ERG5* and *ERG11* mutations exhibit a complete reduction in ergosterol and a cross-resistance to fluconazole and amphotericin B (AmB) (Figure 1A) [10]. *UPC*2 encodes a zinc cluster transcription factor, Upc2, which regulates the uptake and metabolism of sterol. Upc2 and Ndt80 (also a zinc cluster transcription factor) can bind promoters of *ERG* genes [11]. Therefore, overexpressed *UPC2* and *NDT80* will upregulate expressions of *ERG2*, *ERG3*, *ERG4*, *ERG6*, *ERG11,* and *ERG25*, generating fluconazole-resistant *Candida* isolates. On the contrary, the strain susceptibility to fluconazole can be increased by 500 times when *UPC*2 is deleted alone or in combination with *NDT80* (Figure 1A) [11]. Genetic mutations are also associated with cross-resistance. For example, *C. albicans* isolates containing the A516P mutation in *ERG11* exhibit an upregulated *ERG11* expression and a notable cross-resistance to fluconazole, itraconazole, and voriconazole [12].

### 2.2. Genetic Mutations Lead to An Abnormal Efflux Pump

The overexpression of cell membrane transporters can also be responsible for the fungistatic activity of fluconazole. The efficiency of efflux pumps can be increased by the overexpression of *CDR1*/*CDR2* or *MDR*. Therefore, drug molecules are pumped out of the cell, exhibiting enhanced antifungal resistance (Figure 1B). The binding and terminal modification of cis-regulatory elements also affect antifungal resistance. *PDR* genes function during the synthesis of ABC-Ts. The *PDR12* promoter contains a cis-regulatory element for Upc2 binding. Isolates without *UPC2* exhibit abnormal sterols accumulation and membrane permeability, as well as increased susceptibility to fluconazole [13]. While the C-terminal modification of *PDR1* with FLAG tag enables a decreased mRNA concentration by about 50%, the N-terminal modification leads to a 2-fold increase in mRNA concentration. The reduction in *PDR1* leads to a significant reduction in fluconazole resistance. The C-terminal modifications also interfere with the action of Gal11A, which subsequently decrease the expression of *PDR1*, *CDR1*, and *PUP1* [14]. Moreover, the deletion of *FLO8* upregulates the expression of *CDR1/CDR2* and is responsible for increased resistance to various drugs (Figure 1B) [15].

Gain-of-function (GOF) mutations can enable resistance to fluconazole in some species of *Candida*. In some fluconazole-resistant *C. glabrata* isolates, a total of ten synonymous mutations and two missense mutations were identified in *PDR1*, four of which are considered GOF mutations. The corresponding genes are over-transcribed 20 to 40 times and further upregulate the expression of *CDR1* and *PUP1* (*CDR1* upregulated genes), which are responsible for increased fluconazole resistance (Figure 1B) [14,16]. Tac1 and Mrr1 function as transcriptional activators of *CDR1* and *MDR1*, respectively (Figure 1B). GOF mutations of *MRR*1 and *TAC1* can upregulate the expression of *CDR1* and *MDR1*, and increase the MIC of *C. albicans* to fluconazole by 60 to 120 times [17,18].

## 3. Antifungal Resistances Mediated by Aneuploidy

Fungal pathogens have significant genomic plasticity for genomic evolutions and adaptions to environmental perturbations. Aneuploidy is often observed across the *Candida* species and leads to fluconazole resistance. For example, among forty-two detected fluconazole-resistant isolates, twenty-one isolates are aneuploids [19]. While genetic mutations affect protein function, aneuploidies affect gene expression through gene dosage effects [20]. These alterations make strains less susceptible to antimicrobials by affecting drug targets, efflux pumps, and expressions of other genes that are associated with drug resistance [21]. A reported clinical *C. glabrata* aneuploidy isolate, which exhibits a two-fold ergosterol concentration, is resistant to fluconazole. The duplicated chromosome contains the *ERG11* gene, which encodes the target of fluconazole, suggesting that chromosome replication is associated with fluconazole resistance [22,23].

Both whole chromosomes and chromosome segments can be duplicated or deleted, generating aneuploidy. Seventy fluconazole-resistant and fluconazole-susceptible *C. albicans* clinical and laboratory isolates were analyzed by using comparative genomic hybridization (CGH) arrays. The result reveals that the chr5L isochromosome [i(5L)] is the aneuploidy most frequently associated with fluconazole resistance (Figure 2A) [19]. Analyses of expression profiles confirm that the expression of *ERG11* and *TAC1* on chr5L is increased, which contributes to drug resistance and tolerance [19].

Trisomy is also prevalent in fluconazole-resistant *Candida* species. In fluconazole-resistant *C. albicans* isolates, chromosome 4 (chr4) trisomy is identified, on which there are no mutations in resistance-associated genes [24]. Trisomy of chromosome 3 (chr3), chromosome 6 (chr6), and chromosome 7 (chr7) is also detected in fluconazole-resistant isolates [25]. Not surprisingly, most of these chromosomes contain genes that affect azole resistance. For example, chr3 contains *CDR1*, *CDR2,* and *MRR1*. *MRR1* encodes a transcription factor promoting *MDR1* expression (Figure 2B) [26,27,28]. *MDR1* is present on chr6, and *NCP1*, a gene encoding the Erg11 cofactor, is present on chr4 [25].

Aneuploidy can also enable cross-adaptations to unrelated drugs. An isolate of *C. albicans* with chromosome 2 (chr2) trisomy is adapted to both the anticancer drug hydroxyurea and the antifungal drug caspofungin. The cross-adaptation may occur because alleles of two adaptor genes are located on the same chromosome [20]. In fluconazole-resistant *Cryptococcus neoformans* isolates, those with disomy of chromosome 1 (chr1) are adapted to flucytosine (5FC), and those with disomy of both chr1 and chr4 exhibit cross-tolerance to AMB and 5FC [29].

Aneuploidy is not only a readily available variation but also reversible. When exposed to conditions without drugs, resistant aneuploidies could gradually regain their susceptibility to drugs by losing aneuploidy chromosomes. Itraconazole- and fluconazole-resistant *C. albicans* isolates were incubated for more than 150 generations in conditions without drugs. The resistance to itraconazole disappeared, and the resistance to fluconazole decreased to 43% of that of the initial strains [22]. Studies on *C. neoformans* reveal that the loss of aneuploidy is associated with the apoptosis-inducing factor Aif1. The inactivation of the *AIF1* gene enables stable chr1 dimers in the absence of drugs [23].

Polyploids occur during the growth and reproduction of fungi, and different ploidies have different potential to produce genomic changes. Upon exposure to fluconazole, diploid *C. albicans* forms unstable tetraploid intermediates by altering cell cycle progression and undergoing abnormal mitosis [30]. The genome of tetraploid *C. albicans* is more unstable than diploids [31,32]. Tetraploids undergo rapid and dramatic genome reductions, abnormal chromosome segregations, and gene rearrangements, leading to aneuploidy while shrinking to diploids [31].

Indeed, the accuracy of chromosome segregation during mitosis is altered under stressful duress, resulting in karyotypic diversity and promoting adaptive cellular evolution. Fluconazole affects ergosterol biosynthesis and nuclear membrane fluidity, which can further affect the proper nucleus separation. Therefore, normal chromosome segregations are disrupted, leading to the frequent occurrence of fluconazole-resistant polyploids or aneuploids [33]. When exposed to fluconazole and AmB, haploid yeast cells exhibit a high probability of chromosome loss. Similarly, in strains lacking the checkpoint gene *MAD2*, aneuploid cells are frequently detected, and these isolates exhibit fluconazole resistance [34].

## 4. Antifungal Resistances Mediated by LOH

LOH refers to a cross-chromosomal event that results in the loss of entire alleles and the surrounding chromosomal region in a heterozygous cell. LOH is an efficient strategy that allows species to adapt to new environments. In particular, LOH is frequently detected in fluconazole-resistant *C. albicans* isolates [35].

### 4.1. Chromosomal Regions of LOH Associated with Drug Resistance

The diploid genome of *C. albicans* exhibits a high degree of heterozygosity. However, LOH could be identified at specific sites in fluconazole-resistant *C. albicans* isolates [36]. Genomic analysis revealed that these LOHs are the results of recombination events or the loss and duplication of a whole chromosome [26]. Therefore, it can be assumed that the presence of fluconazole may induce genomic rearrangements, which further enhance drug resistance. Unlike aneuploidy, an LOH-associated drug resistance increase is persistent and recurrent [37]. The LOH of chr1, the right arm of chromosome 3 (chr3R), and the left arm of chromosome 5 (chr5L) are identified in clinical isolates, and chr3R and chr5L are statistically recurrent [37].

The chr5 of *C. albicans* is representative of LOH events, on which the genes of *ERG11*, *TAC1*, and the mating-type locus (*MTL* locus) exist [35]. *C. albicans* usually exhibits as a diploid with three different mating types (*MTL***a***/α*, *MTL***a***/***a**, or *MTLα/α*). *MTL***a***/α* diploid cells exhibit an *MTL*-heterozygous biofilm; otherwise, an *MTL*-homozygous biofilm appears. The mating is prevented by *MTL*-heterozygous biofilms, and multiple drug-resistant *C. albicans* strains are homozygous [38]. *TAC1* is located 14 kb from the *MTL* locus, and the LOH of *TAC1* tends to extend to the *MTL* locus. Therefore, strains with homozygous *TAC1* alleles are usually accompanied by homozygous *MTL* loci [39]. The poly(A) polymerase responsible for mRNA adenylation, encoded by *PAP1*, *PAP1-***a***/PAP1-α*, is also adjacent to the *MTL* locus. Homozygous *PAP1-α* alleles enable hyperadenylation and increase the stability of *CDR1* transcripts [40]. Strains with homozygous *PAP1* alleles are also usually accompanied by homozygous *MTL* loci. Further distal to chr5L, LOH of the *EEG11* allele is also often accompanied by homozygous *MTL* loci due to recombination events [38]. Surprisingly, a homozygous mutation in *MRR1* on chr3 results in an increased LOH rate of chr5 [35]. It is assumed that the LOH of *MRR**1* can promote the homozygous *MTL* through the loss of chr5 [38].

### 4.2. Different Types of LOH Associated with Drug Resistance

According to the length of DNA fragments, LOHs are classified into short-range LOH, long-range LOH, or whole-chromosome LOH [33,41]. All three types of LOH are associated with antifungal resistance. Short-range LOHs are also called interstitial LOHs (I-LOHs), which undergo small-scale DNA fragment transfers (<50 kb) (Figure 3A). I-LOH events are the most frequent LOHs and are usually formed by gene conversion [42]. For example, of 96 clinical *C. albicans* isolates analyzed, the majority of drug-resistant strains exhibited homozygous *MTL* loci [43]. The homozygous *MTL* could facilitate the mating and chromosome reassemblies to generate resistant progeny. In a fluconazole-resistant *C. albicans* isolate, a GOF mutation in the *TAC1* allele, N977D, is accompanied by an LOH on chr5 [39]. This LOH event may be due to the gene conversion of *TAC1* after the GOF mutation, leading to an overall enhancement of Tac1 activity [39].

Long-range LOHs are also called terminal LOHs (T-LOHs) with long chromosomal fragments in homozygous LOHs (>50 kb) (Figure 3B). T-LOHs are usually formed by reciprocal cross-over (RCO) or break-induced replication (BIR) events. LOHs in the yeast genome usually correspond to large-scale chromosomal regions containing multiple adjacent genes [44]. Fluconazole-resistant strains usually contain extremely active *TAC1* alleles and homozygous *ERG11* alleles. *TAC1* hyperactivity is generally due to GOF mutations, whereas the homozygosity of *ERG11* is acquired through mitotic recombination events. *ERG11* recombination events occur with the LOH of *TAC1* and *MTL* when the chromosomal breakage is adjacent to the centromere, generating a T-LOH [39]. The *MDR1* gene is located on chr3, and its overexpression is one of the main causes of fluconazole resistance for *C. albicans* [45]. Surprisingly, most fluconazole-resistant isolates exhibit a constitutive upregulation of *MDR1* and homozygous *MRR1* mutations [28]. It reveals that the LOH of *MRR1* can be caused by a mitotic recombination event involving a cross-over between the centromere and the *MRR1* locus. The recombination event results in a T-LOH from the cross-over locus to the telomere [26].

Whole-chromosome LOH (whole-chr LOH) is a phenomenon of chromosome non-disjunction, causing chromosome imbalance in daughter cells. The zygote with one chromatid undergoes a duplication of the remaining chromosome, generating a homozygous diploid [46,47]. The zygote with three chromatids undergoes a loss of the heterozygous homolog to become homozygous (Figure 3C) [47]. Whole-chr LOHs of chr1, chr3, and chr5 are detected in fluconazole-resistant *C. albicans* isolates, which contain numerous fluconazole resistance genes [26,37].

### 4.3. Factors That Could Affect LOHs

When exposed to antifungal drugs, *C. albicans* undergo more genomic recombination than mutations. For instance, when exposed to fluconazole, rates of total LOHs and whole-chr LOHs increase by approximately 285 and 5 times, respectively [33]. The clinical isolates exhibit a higher rate of LOH than laboratory strains. The host-associated strains suffer approximately 10-fold more LOHs than strains that grow in nature [31]. It is assumed that innate immune components, including antimicrobial peptides (AMPs) and reactive oxygen species (ROS), make host-carried *C. albicans* generate more large-scale genomic variations. Meanwhile, chromosomal or segmental aneuploidy is frequently detected in LOH strains [48]. Furthermore, tetraploids endure a more severe drug-induced genomic instability than diploids. When exposed to fluconazole, generated LOHs in tetraploids are 60 times greater than those in diploids [32]. Adaptive mistranslations at CUG sites also affect the rate of LOH events and accelerate the generation of fluconazole-resistant *C. albicans* isolates [49]. In addition, the rate of LOH in strains can be increased by UV and γ-radiation, CRISPR/Cas9, and SCRaMbLE [50,51,52].

## 5. Conclusions

Antifungal resistance is a severe threat to human health. The yeast *Candida* is the most common pathogen, and fluconazole is an efficient therapeutic drug for fungal disease. Unfortunately, fungi can become resistant over time by generating genomic variations when exposed to fluconazole. The resistant fungi are capable of being isolated from the clinical environment and analyzed by genomic analysis. Therefore, we know that fluconazole resistance is related to genomic variations at different levels, including mutations, aneuploidies, and LOHs.

Genetic mutations generate amino acid substitutions, which are closely related to drug resistance. In particular, missense mutations and GOF mutations can alter protein structures or promote the expression of resistance genes. Aneuploidies and LOHs are capable of affecting the copy number of genes and their expression, which further affects drug targets, efflux pumps, and the expression of other drug-resistant genes. Aneuploidy and LOHs are unstable, which can be rapidly acquired under drug stimulation and recovered under drug-free conditions.

In general, the generation of fungal resistance is facilitated by chromosome breakages and recombination. Thus, new therapeutic strategies are being developed to reduce the frequency of antifungal resistance. Agents capable of reducing chromosome breakage and recombination can be used as adjuvants to fluconazole [19]. However, the mechanism of drug resistance is complex and is also related to biofilms, the host environment, and the pharmacokinetics of the drug. To fully elucidate the mechanism of drug resistance, unremitting efforts and powerful research technologies are necessary.

The better we understand these variations, the better we can treat the fungi disease precisely. The development of synthetic biology provides new opportunities for fungal resistant analysis from screening to creation. Through genome synthesis and inducible rearrangement, we can create drug-resistant fungi. The SCRaMbLE technology can accelerate the evolution of synthetic genomes and generate large strain libraries with numerous genomic variants, including aneuploidies, LOHs, and structural variations. It provides an efficient platform for the detection of drug-resistant strains, the analysis of drug resistance mechanisms, and the identification of new drug targets. Furthermore, automated biofoundries and machine learning can be employed to facilitate strain construction and analysis, promoting the understanding of drug-resistant impacts of genomic variations and rational clinical drug use.

## Figures and Tables

**Figure 1 biomolecules-12-00845-f001:**
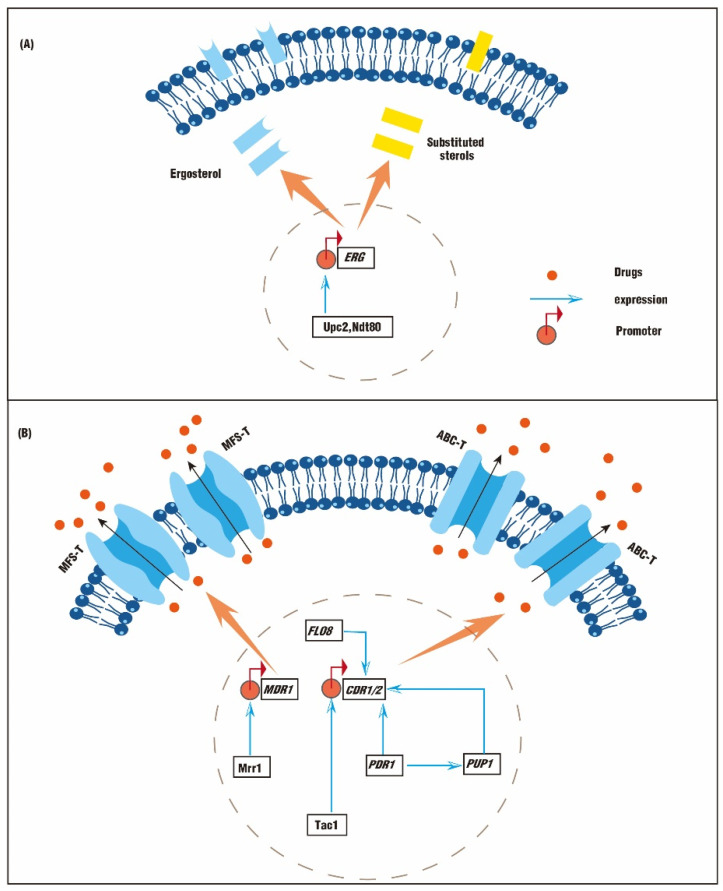
The mechanisms of drug resistance in fungi. (**A**) The resistance is induced by abnormal sterol substitution or accumulation. (**B**) The resistance is induced by abnormal efflux pump function.

**Figure 2 biomolecules-12-00845-f002:**
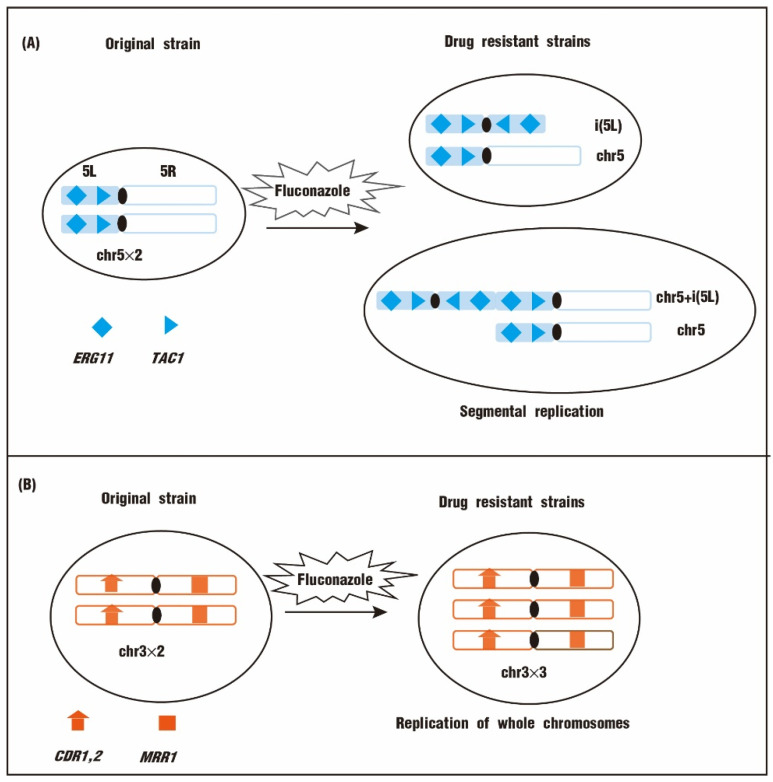
The aneuploidy-mediated fluconazole resistance. (**A**) The segmental aneuploidy i(5L) is related to fluconazole resistance in yeast. (**B**) The trisomy of chr3 is related to fluconazole resistance.

**Figure 3 biomolecules-12-00845-f003:**
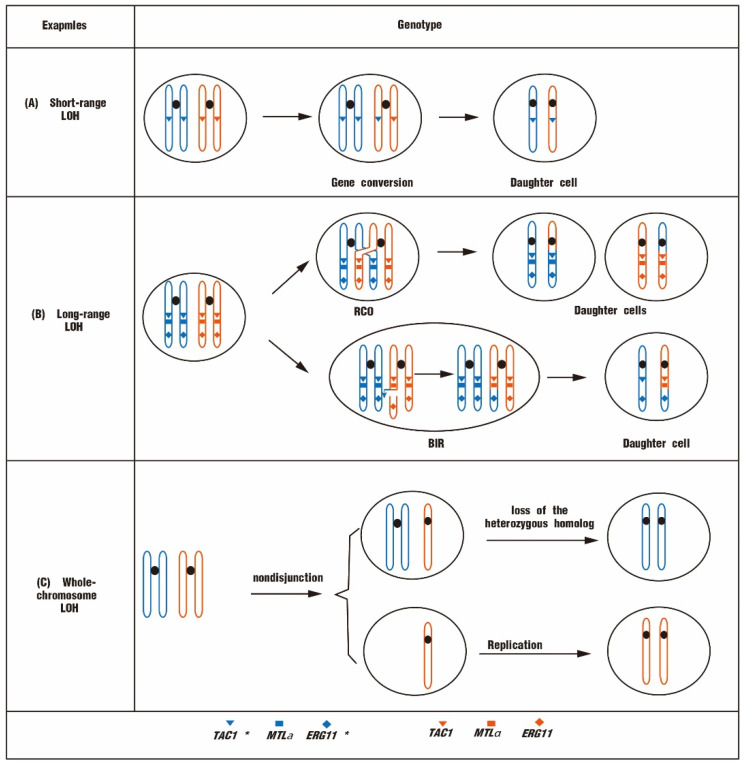
Three different types of LOH are associated with fluconazole resistance in yeast. (**A**) Short-range LOH that is usually produced by gene conversion through drug-resistant mutant genes, leading to an increased copy number of drug-resistant genes. (**B**) Long-range LOH. It results from RCO or BIR events, with intact replication or exchange from the break site to the telomere. (**C**) Whole-chromosome LOH. Nondisjunction or chromosome missegregation events can yield trisomic or monosomic progeny, respectively. Whole-chromosome LOH arises by either loss of the heterozygous homolog in trisomic individuals or reduplication of the hemizygous homolog in monosomic individuals. *TAC1** and *ERG11** represent the genes with mutations.

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
