# Peer review of "Genomic Variation-Mediating Fluconazole Resistance in Yeast"

_biomolecules, 2022, doi:10.3390/biom12060845_

Round 1

Reviewer 1 Report

I believe that this work will contribute to a better understanding of the theme. In my opinion, the manuscript needs a minor revision.  

Comments:

  • Write the complete name of the species when it appears in the text the first time.
  • spp, : is incorrect
  • Line 26: Azoles target ergosterol biosynthetic enzyme lanosterol deme-26 thylase (also referred to as cytochrome P450), encoded by ERG11. In this case, the phrase is meaningless.
  • For reading and understanding the article needs to be an expert on the theme. I suggest that the authors put more information about the genes. In the start, they put but throughout the manuscript, this no occur.
  • Line 40: The synthesis of ergosterol may be affected by ERG The reduction of er-48 gosterol or accumulation of intermediate products may lead to the resistance of Candida 49 to azoles and amphotericin B (AmB). In clinical isolates with ERG mutations, CDR1/CDR2 50 and MDR are overexpressed in varying degrees. Missense mutations in ERG3 are thought 51 to be responsible for the mis-synthesized sterols and high levels of drug resistance. Put the reference
  • Line 70: Cell s, Line 75: in AspERGillus fumigatus, Line 160: Similarly. it is sufficient...revise

Author Response

Dear reviewer,

We would like to thank you and editor for giving us constructive suggestions. Here we submit a new version of our manuscript, modified according to your suggestions. We mark all the changes in the revised manuscript. In addition, changes made according to the reviewers’ comments are outlined as follows. 

Comment 1: Write the complete name of the species when it appears in the text the first time.

Response: The complete name of the species were written when it appears in the text the first time. Please see line 20-21 and 125.

Comment 2: spp, : is incorrect

Response: “C. spp” were deleted. Please see line 21.

Comment 3: Line 26: Azoles target ergosterol biosynthetic enzyme lanosterol deme-26 thylase (also referred to as cytochrome P450), encoded by ERG11. In this case, the phrase is meaningless.

Response: We deleted “(also referred to as cytochrome P450)” and we reorganized this sentence. Please see line 24-27.

Comment 4: For reading and understanding the article needs to be an expert on the theme. I suggest that the authors put more information about the genes. In the start, they put but throughout the manuscript, this no occur.

Response: We added more information about genes. Please see line 38-45, 57-58, 90-91 and 119-121.

Comment 5: Line 40: The synthesis of ergosterol may be affected by ERG The reduction of ergosterol or accumulation of intermediate products may lead to the resistance of Candida to azoles and amphotericin B (AmB). In clinical isolates with ERG mutations, CDR1/CDR2 and MDR are overexpressed in varying degrees. Missense mutations in ERG3 are thought to be responsible for the mis-synthesized sterols and high levels of drug resistance. Put the reference

Response: We rewrote the description about this part and corresponding references were cited. Please see line 49-53.

Comment 6: Line 70: Cell s, Line 75: in AspERGillus fumigatus, Line 160: Similarly. it is sufficient...revise

Response: We deleted the paragraph with “Cell s” and “in AspERGillus fumigatus”. Please see 66-47. “Similarly. it is sufficient...” was revised, please see line 147-149.

Reviewer 2 Report

Major comments

  1. The authors use the term "azoles". This is a kind of unacceptable slang. I may suspect that this term may cover the first generation drugs, like ketoconazole, which are indeed azole derivatives. but also the second generation fluconazole or voriconazole, which are the triazole derivatives. It should be clearly defined in the first part of the ms., which particular drugs are covered by the general term.
  2. I would expect that in the review paper on fungal resistance to azole and triazole antifungals, the particular mutations in ERG genes, especially ERG 11, would be specified in detail. The same may also apply to mutations in other genes
  3. Although most of the mutations responsible for fungal resistance to "azole" antifungals indeed induce resistance to all compounds of this type, there are also examples of mutations, especially in ERG 11, which cause resistance to some, particular compounds of this group but not to the other ones. This issue is not addressed in this ms.
  4. Language errors must be corrected. There are several typographic and grammar errors throughout the body text. Extensive correction by any native English person is recommended.

Minor comment

lines 26-27

lanosterol demethylase is not referred to as cytochrome P450. The enzyme, lanosterol demethylase, is cytochrome P450 dependent.

Author Response

Dear reviewer,

We would like to thank you and editor for giving us constructive suggestions. Here we submit a new version of our manuscript, modified according to your suggestions. We mark all the changes in the revised manuscript. In addition, changes made according to the reviewers’ comments are outlined as follows.

Comment 1: The authors use the term "azoles". This is a kind of unacceptable slang. I may suspect that this term may cover the first generation drugs, like ketoconazole, which are indeed azole derivatives. but also the second generation fluconazole or voriconazole, which are the triazole derivatives. It should be clearly defined in the first part of the ms., which particular drugs are covered by the general term.

Response: This manuscript mainly reviews the resistance of fluconazole, so we narrowed down the concept of azoles to fluconazole.

Comment 2: I would expect that in the review paper on fungal resistance to azole and triazole antifungals, the particular mutations in ERG genes, especially ERG11, would be specified in detail. The same may also apply to mutations in other genes

Response: We added detailed description related to mutations in ERG11 and other genes. Please see line 49-50, 53-55, 64-66 and 186-187.

Comment 3: Although most of the mutations responsible for fungal resistance to "azole" antifungals indeed induce resistance to all compounds of this type, there are also examples of mutations, especially in ERG11, which cause resistance to some, particular compounds of this group but not to the other ones. This issue is not addressed in this ms.

Response: We narrowed down the "azole" antifungals to “fluconazole” antifungals, and reviewed functional mutations that are responsible for fluconazole resistance. All corresponding changes were marked in the manuscript, such as line 49-52, 53-55, 64-66 and 91-92.

Comment 4: Language errors must be corrected. There are several typographic and grammar errors throughout the body text. Extensive correction by any native English person is recommended.

Response:We revised language errors throughout the body text and all changes were marked.

Comment 5: lines 26-27 lanosterol demethylase is not referred to as cytochrome P450. The enzyme, lanosterol demethylase, is cytochrome P450 dependent.

Response: We deleted “(also referred to as cytochrome P450)” and we reorganized this sentence. Please see line 24-27.

Round 2

Reviewer 2 Report

The authors have adequately addresssed all comments. I would suggest acceptance of the revised version of the ms.

Author Response

Dear reviewer,                                      

We would like to thank you for giving us constructive suggestions. Here we submit a new version of our manuscript, modified according to your suggestions. We proofread the manuscript and mark all the changes in the revised manuscript.

Thank you very much for your kindness and help again! We would be glad to provide any additional information if needed, please feel free to contact us.

Yours sincerely,

Ze-Xiong Xie
Associate Professor of Pharmaceutical Engineering
Frontier Science Center for Synthetic Biology and Key Laboratory of Systems Bioengineering (Ministry of Education)
School of Chemical Engineering and Technology
Tianjin University, Tianjin 300072, China
Tel: +86-15022512256
E-mail: [email protected]